# The epidemiological risk factors of hand, foot, mouth disease among children in Singapore: A retrospective case-control study

Jo Ann Kua[1], Junxiong Pang[1,2]*

1 Saw Swee Hock School of Public Health, National University of Singapore and National University Health System, Singapore, Singapore, 2 Centre for Infectious Disease Epidemiology and Research, National University of Singapore, Singapore, Singapore

* pangv@hotmail.com

## Abstract

The incidence of hand, foot, and mouth disease (HFMD) is increasing over the years despite current prevention and control policies in Singapore. A retrospective case-control study was conducted among parents whose children attended childcare centres in Singapore to assess the epidemiological risk factors associated with HFMD among children below 7 years old. Parents of 363 children with HFMD (as cases) and 362 children without HFMD (as controls) were enrolled from 22 childcare centres. Data of potential risk factors were collected through a standardised self-administered questionnaire from parents which include demographics and hygiene practices. Multivariate analysis were adjusted for age group, parent's education level, mother's age, HFMD-infected siblings, and preschool admission period. Child's age between 1.5 and 4.9 years, child who had been in childcare for more than 1.9years, having HFMD-infected siblings, two or more children in a family, higher educated parents, parents who had HFMD episode previously, wash toys with soap once every two to three weeks, sanitise toys once every two to three weeks, out-sourced cleaner in childcare centre, no domestic helper at home and more than 22 children in a classroom were independent risk factors of HFMD. These evidence provide crucial implications to guide more effective prevention and control of HFMD in Singapore.

## Introduction

### Clinical features and virology of HFMD

Hand, foot and mouth disease (HFMD) is a prevalent infection among children in East and Southeast Asia caused by a group of human enteroviruses from the family *Picornaviridae*. More than 20 serotypes of enteroviruses can cause HFMD and the most common etiological agents include Enterovirus A71 (EV-A71), Coxsackievirus A16 (CV-A16) and Coxsackievirus A6 (CV-A6) [1, 2]. HFMD is usually mild and self-limiting with symptoms include >38˚C fever, vesicular rash on the palms, soles, tongue or buttocks, sore throat, and ulcers at the front of the mouth. Herpangina (HA) is one of the clinical manifestation of HFMD which is caused

**Funding:** JAK-Singapore Children's Society (https://www.childrensociety.org.sg/research-grants) JAK- Saw Swee Hock School of Public Health (https://sph.nus.edu.sg/) The funders had no role in study design, data collection and analysis, decision to publish, or preparation of the manuscript.

**Competing interests:** The authors have declared that no competing interests exist.

by the same group of enteroviruses, except that the clinical diagnosis of HA is characterised by oral blisters on the roof of the mouth and at the back of the throat without vesicular rash [3]. There were no significant differences found in age, gender, circulating season, and the diversity of viral genome between HFMD and HA infected individuals [4]. Among all viruses, EV-A71 infection is prone to severe complications which can lead to brainstem encephalitis or pulmonary oedema [5]. Fatal HFMD outbreaks hadled to many deaths in many countries including Singapore, Malaysia, Vietnam, Cambodia, Taiwan, and China [6–11].

## Transmission of HFMD

HFMD can transmit through faecal-oral route, direct contact with respiratory droplets, nasal discharge, saliva, vesicularfluids, or articles/fomites contaminated by secretions from an infected person. The incubation period is usually 3 to 5 days but can be up to 2 weeks. HFMD can be asymptomatic but contagious as the infectious period starts a few days to about a week before onset of illness. The duration of virus shedding often depends on the type of infected strain and severity of illness [12]. Prolonged viral shedding of EV-A71 virus can last up to 4 weeks in throat and persist for 6–12 weeks in faeces excretion [13, 14]. Majority of HFMD cases occur at preschool aged below 5 years with the highest age-specific annual incidence rate for 0to4 years old but adults can be affected sporadically [6, 15]. The basic reproduction number ($R_0$) was estimated to range from 2.4 to 5.5 for different serotypes based on studies in Singapore and Hong Kong [16, 17]. A person recovered from infection by a particular enterovirus serotype develops protective immunity against that specific viral serotype but can still be infected again by different serotypes of enteroviruses. HFMD outbreaks were found to be seasonal in temperate (incidence peak during summer) and subtropical Asia (incidence peak during late spring and fall), but no clear pattern in tropical Asia [18]. However based on a systematic and meta-review, the meteorological effect was weak with no evidence of strong relationship between humidity, temperature and HFMD transmission [18].

## HFMD in Singapore

HFMD was first detected in June to July 1970 in Singapore [19]. Since then, outbreaks caused by CV-A16 [19] and EV-A71 [20] were reported. CV-A16 was the main circulating enterovirus in 2005, 2007 and 2009, and EV-A71 in 2006 and 2008 [21, 22]. Subsequently after late 2009, CV-A6 became the predominant circulating virus associated with HFMD outbreaks [2]. It was found that HFMD mainly affected infants and young children aged 0 to 4 years old of both gender in Singapore [21, 23]. The male to female ratio of 1.2: 1 was the same in 1981 and 2017 [15, 19]. However, males were predominant during HFMD outbreak in 2000. The highest ethnic-specific incidence rate had shifted from Chinese in 1981 to Malay in 2017 [15, 19]. A study on risk factors for fatal or severe HFMD cases was conducted in Singapore [5]. A case-control study during an outbreak in 1981 discovered that household contacts and sharing of household articles were important risk factors of HFMD transmission [19]. The first cross-sectional serologic study in Singapore published in 2002 reported that HFMD infected preschool-aged children the most and found that preschool setting such as concentration of susceptible population in a confined space (classroom) and sharing of toys contributed to spreading of HFMD infections [24]. Nevertheless, very few studies focused on risk factors and protective factors for prevention of HFMD in Singapore.

## Risk factors of HFMD

Based on a systematic review for risks factors of HFMD, it was suggested that young age of less than five, male gender, poor hygiene, and high frequency of social contacts are associated with

high risk of HFMD infections [18]. About 82% of HFMD cases in Asia occurred before 6 years old [18] while the presence of maternal antibodies could be a reason of lower incidence rate before 1 year of age [24]. Male was found to be a risk factor of HFMD in most literatures [20, 21, 25–28], while a few studies shown borderline evidence due to gender differences [29, 30]. Contact with a case especially within the household is a significant risk factor of HFMD transmission [19, 29, 31]. Although some studies suggested that attended preschool was associated with increased risk of HFMD [21, 32], a case-control study in China shown that other risk factors include close contact with neighbourhood children and exposed to crowded places had greater risk than attended preschool [33]. Contrariwise, a case-control/case-crossover study in Korea found that household size and attended preschool were not associated with the risk of HFMD [34].

## Protective factors of HFMD

A frequency-matched case-control study in China found that good hand-washing habits which include washing hands with soap, children always wash hands before meals, and adult always wash hands before feeding child, were protective against HFMD with a risk reduction of >95% during outbreaks [33]. A cluster-randomized controlled trial on hand hygiene intervention among kindergartens in China shown to be effective at reducing HFMD in children [35]. A community intervention study in China suggested that community education on hand washing can improve personal hygiene and prevent HFMD transmission [36]. A retrospective case-control study in China suggested that exclusive breastfeeding can prevent HFMD infections for the first 28 months of age compared to mixed feeding [37]. Li et al. discovered that prolonged breastfeeding for 6 months and above was significantly associated with lower risk of severe HFMD [38]. Another study also found that breastfeeding was protective against severe HFMD but the duration of breastfeeding was not specified [28]. Zhu et al. reported that prolonged exclusive breastfeeding for at least 4 months and higher gestational age was associated with lower incidence of fever in HFMD infections [39].

## Vaccination of HFMD

Frequent variations in the viral genome of enteroviruses and lack of evidence on cross-protection from different strains remains a challenge in the development of effective vaccines for HFMD [40]. Inactivated monovalent EV-A71 vaccines have been approved in China with supporting studies proven that the vaccine can prevent more than 90% of EV-A71 infections and symptoms [41]. However, EV-A71 vaccination may lead to an increase in HFMD caused by other co-circulating serotypes during epidemics [40]. As there is no multivalent vaccine currently available for HFMD that targets all prevalent serotypes of enteroviruses, effective preventive measures to target risk factors are crucial to control the transmission.

## Current challenges

National surveillance on HFMD was implemented as one of the control measures in Singapore to monitor this public health issue. Since October 2000, HFMD became a legally notifiable disease to Ministry of Health (MOH). The incidence of HFMD is relatively high compared to other notifiable infectious diseases over the years despite current control policies include childcare centre surveillance, case-isolation, publishing names and short-term mandatory closure of childcare centre with active clusters of prolonged transmission. According to the weekly infectious disease bulletin published by the Communicable Diseases Division of MOH, a total of 40,217 cases of HFMD were reported in 2018 which was 19.47% higher compared to 33,663 cases reported in2017 while the year 2016 had the highest record of 42,154 cases ever since

year 2012 (37,125 cases) [42]. A recent paper researched on the impact of HFMD control policy in Singapore highlighted the gaps of current control policies and suggested that the focus should move towards preventive guidelines and education to reduce HFMD infections in Singapore [43].

Studies in Singapore had been focusing on the disease trend, virology, serology and clinical features of HFMD since the outbreak in late September 1972. Nonetheless, there were limited well-powered studies on risk factors and protective factors of HFMD in Singapore. Although studies in China found proper hand washing with soap reduced HFMD infections, no local evidence was found to validate the approach of proper hand washing with soap for the prevention of HFMD transmission in Singapore. Furthermore, limited studies in China and none in Singapore were identified for breastfeeding as a potential protective factor of HFMD infections. Hence, more research on epidemiological risk factors is required to fill the gaps of inadequate evidence to guide in the development of evidence-based interventions and policy to reduce the burden of HFMD in Singapore.

This study aimed to provide evidence-based guidance for more effective prevention and control of HFMD infections in Singapore. The objective was to identify epidemiological risk factors and protective factors associated with HFMD infections in preschool children age 6 and below. This study hypothesized that male, contacts with HFMD-infected siblings, siblings attending same childcare centre, longer preschool admission, plays in public playgrounds, and sharing utensils with siblings were risk factors of HFMD infections while exclusive breastfeeding, higher gestational age, and frequent hand washing were protective against HFMD infections in children.

## Methods

### Study design and study population

A retrospective unmatched case-control study was conducted to evaluate the differences in exposures of interest between cases and controls in order to determine the potential risk and protective factors associated with HFMD infections in children. Study population was the parents of preschool-age children who attended childcare centre in Singapore from year 2016 to 2018. Parents of cases and controls were recruited from 22 childcare centres.

### Definition of case and control

A case was defined as a child who was clinically diagnosed with HFMD between year 2016 and 2018. A control was defined as a child with no history of clinically diagnosed HFMD between year 2016 and 2018. Clinical diagnosis of HFMD were extracted retrospectively from de-identified clinician-endorsed medical certificate records provided by the childcare centres. Cases and controls were systematically identified from all participating childcare centres who were enrolled between 2016 and 2018 before the recruitment of their parents to do the questionnaire.

### Recruitment

The research protocol with waiver of written informed consent was approved by the National University of Singapore (NUS) Institutional Review Board (IRB). Upon IRB approval, participated childcare centres were approached for recruitment of the parents of cases and controls. All methods were performed in accordance with the relevant guidelines and regulations as mandated by NUS IRB. Personal identifiers were not collected throughout the research. Participants were anonymized with a study identification (ID) number encrypted with a

combination of alphabets and numbers. The dissemination of hardcopy study documents to parents including participant information sheet and standardised self-administered questionnaire to cases and controls was conducted by respective childcare centres. By returning the completed questionnaire to childcare centres, participants were deemed to have given consent to participate in the study. Verbal and email reminders were given to participants who did not response within a week after the dissemination of questionnaire. Completed questionnaires were collected from childcare centres two weeks later. Participants were not followed-up after completing the questionnaire. Withdrawal of participation from the study was allowed at any point of the study with a written note or email to the study team. Information collected will be removed from the dataset after withdrawal from the study. Recruitment of participants was completed in January 2019.

## Questionnaire design

Exposures of interest were collected in a standardized manner for both cases and controls via self-administered questionnaire by the parents that can be completed in less than 20 minutes. The questionnaire was designed with the hypotheses in mind to gather sufficient information for analysis (Supplementary Information). Close-ended questions were used to maximise responses for all items and to ease comprehension by participants. The items in questionnaire were arranged according to different sections for demographics, health information, risk factors and knowledge of HFMD. Screening questions were included to exclude participants who were not eligible for the study. The questionnaire was developed in English language and then translated into Chinese language, vetted by a bilingual Chinese Curriculum Specialist using forward and back translations to achieve linguistic equivalence between both versions. Both versions were pre-tested by a convenience sample of respondents from a similar target population to improve the questionnaire and to ensure appropriateness of response options prior to official administration.

## Data collection

Data collected for both cases and controls include demographics, social contacts, health characteristics, maternal, birth and infancy factors, hygiene practices, and knowledge of HFMD. The characteristics of childcare centres were collected from participated childcare centres which include region, building type, classroom layout, playground, cleaner, ventilation, enrolment (total children), class size, and hand sanitiser in classroom. Preschool admission period in years was derived from the duration between date of admission into childcare centre and date of recruitment into the study. Age of disease onset in years was calculated based on the start date of medical certificate record and date of birth. Recovery days was calculated based on the start and end date of medical certificate record and/or fit for school certificate from clinical consultation. Knowledge of HFMD was calculated as mean of total score obtained from 15 questions in the questionnaire. A correct response for each knowledge question was scored "1" and an incorrect or indeterminate response was scored "0". Proportion of correct responses for each question was tabulated for both cases and controls.

## Statistical analysis

Data collected were coded into numbers and analysed statistically. Continuous data were summarised as means ± standard deviations (SD). Student's t test was used to test for differences in the means of continuous variables with normally distributed data. Categorical data were summarised as numbers and percentages. Chi-squared test was used to test for differences in the proportions of categorical variables. Chi-square linear test was used to test for linear trend in

ordinal categories of proportion. Unconditional logistic regression analysis was performed to assess the association of potential risk factors and HFMD. All factors were tested with the univariate logistic analysis and all significant variables were included in the multivariate logistic regression analysis. Univariate logistic regression model was used to report unadjusted odds ratios (ORs) of each independent variable. Adjusted ORs were derived from the multivariate logistic regression model controlled for age group, parent's education level, mother's age, HFMD-infected siblings, and preschool admission period as potential confounders. ORs were reported with 95% confidence intervals (CI) and two tailed $p$-values at a significant level of $p <$ 0.05. Statistical analysis was performed using STATA statistical software package Version 15.1 (Stata Corp LP, College Station, TX).

## Results

### Demographics and health characteristics

Parents of a total of 363 cases and 362 controls were recruited from 22 childcare centres, among 2,180 children systematically identified, which had given consent to participate in the study. The response rate is about 33.3%. Among these cases, 12 cases (3.3%) were hospitalized due to HFMD. Significant difference was observed for age distribution between cases and controls ($p <$ 0.001; Table 1). Cases had on average 0.4 years longer period of preschool since first admission than controls (95% CI 0.3 to 0.6, $p <$ 0.001). No statistical differences were found in gender and ethnicity among case and control group while nationality was at borderline significance difference.

There were significantly more siblings in the same childcare centre among the cases than controls ($p =$ 0.001). The proportion of siblings previously infected with HFMD was significantly higher among cases compared to controls ($p <$ 0.001). More cases than controls had rotavirus vaccination ($p =$ 0.020). The mother's age among the cases was significantly younger than the controls ($p =$ 0.029). There were more cases with more than one children in the family compared to controls ($p =$ 0.045). The household size was greater among the cases than in the controls($p =$ 0.016). There were significantly more parents with tertiary or higher level of education among cases than controls ($p =$ 0.009). Differences in housing type, housing space and household monthly income were not significant between cases and controls.

More cases than controls had either one or both parents with a history of HFMD compared to controls ($p <$ 0.001). Grandparents were more likely to be the main caretaker of sick child among cases than controls ($p =$ 0.002). Among the cases, 43.2% of parents had spent an average medical costs of more than S\$100 and 39.3% had spent between S\$51 and S\$100 to seek medical treatment for HFMD. Only 2% of parents had the medical costs covered by company medical insurance and 98% had to pay these medical fees out-of-pockets. 47.1% of parents took an average of 2 to 4 days of work leave and 38.5% took more than 4 days to take care of their children with HFMD.

No statistical differences were observed for cases and controls in terms of birth weight, gestational age, mode of delivery, mother's age at child birth, breastfeeding and age of starting solids (weaning age) (Table 2).

### Risk factors and knowledge of HFMD

Cases were significantly more likely to play in outdoor playground as compared to controls ($p =$ 0.004; Table 3). However, no statistical differences were found for child plays in indoor playground, child plays with neighbourhood children, child's frequent hand washing with soap (before eating, after eating, after toilet), adult's frequent hand washing with soap (after diaper changing or washing up child after toilet, before feeding child), frequent use of hand

**Table 1. Demographics and health characteristics.**

| | Cases | Controls | p-value* |
|---|---|---|---|
| | (n = 363) | (n = 362) | |
| Age at recruitment (years) (mean ± SD) | **3.9 ± 1.3** | 3.5 ± 1.6 | < 0.001 |
| Age group at recruitment (%) | | | < 0.001 |
| 0.4–1.5 years | 12 (3.3) | 50 (13.8) | |
| 1.5–2.9 years | 82 (22.6) | 101 (27.9) | |
| 3–3.9 years | 105 (28.9) | 68 (18.8) | |
| 4–4.9 years | 88 (24.2) | 59 (16.3) | |
| 5–5.9 years | 59 (16.3) | 63 (17.4) | |
| 6–6.9 years | 17 (4.7) | 21 (5.8) | |
| Preschool admission period (years) (mean ± SD) | 1.9 ± 1.1 | 1.5 ± 1.1 | < 0.001 |
| Infected once[i] | 1.8 ± 1.0 | 1.5 ± 1.1 | 0.003 |
| Infected more than once[ii] | 2.1 ± 1.0 | 1.5 ± 1.1 | < 0.001 |
| Age of disease onset (years)[a] (mean ± SD) | | | |
| First HFMD reported[1] | 2.5 ± 1.1 | | |
| Second HFMD reported[2] | 3.1 ± 1.1 | | |
| Recovery days[b] (mean ± SD) | | | |
| First HFMD reported[1] | 7.4 ± 2.7 | | |
| Second HFMD reported[2] | 7.8 ± 2.8 | | |
| Gender (%) | | | 0.148 |
| Male | 196 (54.0) | 176 (48.6) | |
| Female | 167 (46.0) | 186 (51.4) | |
| Nationality (%) | | | 0.049 |
| Singaporean | 352 (97.0) | 340 (93.9) | |
| Non-Singaporean | 11 (3.0) | 22 (6.1) | |
| Ethnicity (%) | | | 0.214 |
| Chinese | 299 (82.4) | 278 (76.8) | |
| Malay | 44 (12.1) | 51 (14.1) | |
| Indian | 7 (1.9) | 12 (3.3) | |
| Others | 13 (3.6) | 21 (5.8) | |
| Siblings in same childcare centre (%) | | | 0.001 |
| 0 | 196 (54.0) | 245 (67.7) | |
| 1 | 142 (39.1) | 101 (27.9) | |
| 2–3 | 25 (6.9) | 16 (4.4) | |
| Sibling's HFMD infection history (%) ^ | | | < 0.001 |
| Yes | 176 (48.5) | 79 (21.8) | |
| Flu in the last 3 months (%) | 242 (66.7) | 234 (64.6) | 0.566 |
| Diarrheal in the last 3 months (%) | 48 (13.2) | 51 (14.1) | 0.734 |
| History of chicken pox (%) | 30 (8.3) | 15 (4.1) | 0.065 |
| History of eczema (%) | 34 (9.4) | 29 (8.0) | 0.803 |
| Optional rotavirus vaccination[c] (%) | 261 (72.1) | 220 (64.0) | 0.020 |
| Mother's age (%) | | | 0.029 |
| 21–30 years | 61 (16.9) | 62 (17.2) | |
| 31–40 years | 278 (76.8) | 255 (70.8) | |
| 41–50 years | 23 (6.3) | 43 (12.0) | |
| Father's age (%) | | | 0.144 |
| 21–30 years | 24 (6.7) | 25 (7.0) | |
| 31–40 years | 259 (72.8) | 237 (66.4) | |

(*Continued*)

**Table 1.** (*Continued*)

| | Cases | Controls | *p*-value* |
|---|---|---|---|
| | (n = 363) | (n = 362) | |
| 41–60 years | 73 (20.5) | 95 (26.6) | |
| Total children (%) | | | **0.049** |
| 1 | **94 (25.9)** | **124 (34.2)** | |
| 2 | **202 (55.6)** | **178 (49.2)** | |
| > 2 | **67 (18.5)** | **60 (16.6)** | |
| Household size (%) | | | **0.016** |
| 2 | **2 (0.5)** | **3 (0.8)** | |
| 3 | **60 (16.5)** | **94 (26.0)** | |
| 4 | **149 (41.1)** | **125 (34.5)** | |
| > 4 | **152 (41.9)** | **140 (38.7)** | |
| Household members (%) | | | **0.027** |
| Parents and children | **192 (56.9)** | **206 (56.9)** | |
| Parents, children and domestic helper | **88 (24.2)** | **78 (21.6)** | |
| Grandparents, parents and children | **65 (17.9)** | **45 (12.4)** | |
| Grandparents, parents, children and domestic helper | **18 (4.9)** | **33 (9.1)** | |
| Highest education of parents (%) | | | **0.009** |
| Non-tertiary | **36 (10.0)** | **60 (16.7)** | |
| Tertiary & above | **324 (90.0)** | **300 (83.3)** | |
| Housing type (%) | | | 0.343 |
| Public housing | 293 (80.7) | 286 (79.2) | |
| Private housing | 70 (19.3) | 73 (20.2) | |
| Housing space (%) | | | 0.105 |
| 1 to 3 rooms | 90 (24.9) | 71 (19.8) | |
| > 3 rooms | 272 (75.1) | 287 (80.2) | |
| Household monthly income (%) | | | 0.486 |
| < S$8,000 | 195 (54.8) | 189 (53.2) | |
| S$8,000 –S$10,000 | 65 (18.2) | 57 (16.1) | |
| > S$10,000 | 96 (27.0) | 109 (30.7) | |
| Parents with diabetes (%) | | | 0.629 |
| None | 345 (95.0) | 349 (96.4) | |
| Mother or father | 16 (4.4) | 12 (3.3) | |
| Both | 2 (0.6) | 1 (0.3) | |
| Parents with hypertension (%) | | | 0.442 |
| None | 342 (95.3) | 338 (93.1) | |
| Mother or father | 18 (4.9) | 23 (6.3) | |
| Both | 3 (0.8) | 1 (0.3) | |
| Hyperlipidaemia parents (%) | | | 0.331 |
| None | 352 (97.0) | 348 (96.1) | |
| Mother or father | 8 (2.2) | 13 (3.6) | |
| Both | 3 (0.8) | 1 (0.3) | |
| Parents with historical HFMD(%) | | | **< 0.001** |
| None | **287 (79.1)** | **338 (93.4)** | |
| Mother or father | **66 (18.2)** | **20 (5.5)** | |
| Both | **10 (2.7)** | **4 (1.1)** | |
| Healthcare provider (%) | | | |
| Private GP | 226 (62.3) | 201 (55.5) | 0.065 |

(*Continued*)

**Table 1.** (Continued)

| | Cases | Controls | p-value* |
|---|---|---|---|
| | **(n = 363)** | **(n = 362)** | |
| Private PD | 154 (42.4) | 164 (45.3) | 0.435 |
| Private hospital | **20 (5.5)** | **38 (10.5)** | **0.013** |
| Polyclinic | **57 (15.7)** | **116 (32.0)** | **< 0.001** |
| Public hospital | **42 (11.6)** | **78 (21.6)** | **< 0.001** |
| Main caretaker when sick (%) | | | |
| Parents | 291 (80.2) | 295 (81.5) | 0.650 |
| Grandparents | **141 (38.8)** | **101 (27.9)** | **0.002** |
| Domestic helper | 64 (17.6) | 62 (17.1) | 0.858 |
| Average work leave taken due to child's HFMD (%) | | | |
| 0 day | 32 (8.9) | | |
| 1 day | 20 (5.5) | | |
| 2–4 days | 170 (47.1) | | |
| > 4 days | 139 (38.5) | | |
| Average medical cost due to child's HFMD (%) | | | |
| S$0 | 7 (2.0) | | |
| < S$50 | 56 (15.5) | | |
| S$50 –S$100 | 142 (39.3) | | |
| > S$100 | 156 (43.2) | | |

Values are means ± SD or n (%).

* Student's t-tests or chi-square tests.

^ At the point of performing the questionnaire.

[i] Derived from 205 cases.

[ii] Derived from 158 cases.

[a] Calculated based on start date of medical certificate and date of birth.

[b] Calculated based on start and end date of medical certificate and/or fit for school certificate.

[c] Account for children age > 8 months old.

[1] Derived from 236 observations.

[2] Derived from 97 observations.

SD, standard deviation; HFMD, hand, foot and mouth disease; GP, general practitioner; PD, paediatrician.

sanitiser for child and frequent use of sanitiser for baby seats. There were more controls whose toys were washed more frequently (at least once a week) than cases. Similarly, more controls sanitised toys more frequently (at least once a week) than controls. Household cleaning frequency did not differ between case and control group. There were more cases who shared utensils with siblings at home compared to controls ($p = 0.04$).

HFMD knowledge was scored based on 15 questions about the transmission route, signs and symptoms, incubation period, susceptible population, treatment, vaccine, prevention and control measures, and viral shedding. Mean score for cases was higher than controls ($p = 0.002$). More cases answered correctly for 13 out of the 15 questions. There were 5 questions with significantly higher score among the cases compared to controls: i) Question 2: Fever, mouth ulcers, rash or blisters on palms, soles, and/or buttocks are some common signs and symptoms of HFMD ($p = 0.033$); ii) Question 3: Incubation period (period from infection to onset of symptoms) of HFMD is usually 3 to 5 days and range from 2 days to 2 weeks ($p = 0.01$); iii) Question 4: Adults can get infected with HFMD ($p = 0.001$); iv) Question 8:

**Table 2. Maternal, birth and infancy factors.**

| | Cases | Controls | p-value* |
|---|---|---|---|
| | (n = 363) | (n = 362) | |
| Birth weight (%) | | | 0.530 |
| < 2.5 kg | 30 (8.3) | 30 (8.4) | |
| 2.5–3.4 kg | 253 (70.1) | 261 (73.3) | |
| > 3.4 kg | 78 (21.6) | 65 (18.3) | |
| Gestational age (%) | | | 0.256 |
| < 37 weeks | 28 (7.8) | 37 (10.3) | |
| 37–42 weeks | 329 (92.2) | 323 (89.7) | |
| Mode of delivery (%) | | | 0.878 |
| Natural birth | 222 (61.7) | 220 (61.1) | |
| Caesarean section | 138 (38.3) | 140 (38.9) | |
| Mother's age at child birth (%) | | | 0.192 |
| < 29 years old | 136 (37.7) | 118 (32.6) | |
| 30–34 years old | 154 (42.6) | 155 (42.8) | |
| > 34 years old | 71 (19.7) | 89 (24.6) | |
| Breastfeeding (%) | | | 0.381 |
| Never breastfeed | 18 (5.0) | 27 (7.5) | |
| Mixed breastfeed with infant formula | 181 (50.0) | 175 (48.3) | |
| Exclusive breastfeed for 2 months or more | 163 (45.0) | 160 (44.2) | |
| Age of starting solids (%) | | | 0.103 |
| 4–6 months old | 151 (41.8) | 126 (35.4) | |
| 7–9 months old | 173 (47.9) | 179 (50.3) | |
| 10 months old & above | 37 (10.3) | 51 (14.3) | |

Values are n (%).

* Chi-square tests.

There is no specific treatment for HFMD besides relief of symptoms ($p < 0.001$); v) Question 9: There is no HFMD vaccine currently available ($p < 0.001$).

## Characteristics of childcare centre

No statistical differences among cases and controls were reported for region, building type, classroom layout, playground, ventilation, class size, and classroom with hand sanitiser of childcare centres (Table 4). The total number of children enrolled in the participating childcare centres was significantly different among cases and controls. More controls were from childcare centres with 51–86 children, while more cases were from childcare centre with 116–260 children. Significant difference was observed for cleaner where more cases were from childcare centres with out-sourced cleaner, while more controls were from childcare centres with in-house cleaner ($p = 0.001$).

## Association between HFMD and risk factors

The association between HFMD and risk factors was reported based on univariate and multivariate logistic regression model adjusted for age group, parent's education level, mother's age, HFMD-infectedsiblings, and preschool admission period. Nationality ($p = 0.049$), siblings in same childcare centre ($p = 0.001$) and household size ($p = 0.016$) were not included in the

**Table 3. Risk factors and knowledge of HFMD.**

| | Cases | Controls | *p*-value* |
|---|---|---|---|
| | **(n = 363)** | **(n = 362)** | |
| Plays in outdoor playground (%) | | | **0.004** |
| Never | **25 (6.9)** | **48 (13.3)** | |
| Once every 2 to 3 weeks | **130 (35.8)** | **94 (25.9)** | |
| Once a week | **95 (26.2)** | **102 (28.2)** | |
| More than once a week | **113 (31.1)** | **118 (32.6)** | |
| Plays in indoor playground (%) | | | 0.065 |
| Never | 68 (18.7) | 82 (22.7) | |
| Once every 2 to 3 weeks | 228 (62.8) | 192 (53.0) | |
| Once a week | 43 (11.9) | 56 (15.5) | |
| More than once a week | 24 (6.6) | 32 (8.8) | |
| Plays with other children in the neighbourhood (%) | | | 0.515 |
| Never | 136 (37.5) | 136 (37.5) | |
| Once every 2 to 3 weeks | 89 (24.5) | 81 (22.4) | |
| Once a week | 54 (14.9) | 68 (18.8) | |
| More than once a week | 84 (23.1) | 77 (21.3) | |
| Child **always** washes hands with soap before eating (%)[a] | 147 (42.4) | 149 (48.4) | 0.123 |
| Child **always** washes hands with soap after eating (%)[a] | 138 (39.8) | 125 (40.7) | 0.805 |
| Child **always** washes hands with soap after toilet (%)[a] | 194 (57.7) | 162 (54.9) | 0.476 |
| Adult **always** washes hands with soap after changing diaper or washing up child after toilet (%) | 296 (81.5) | 303 (83.7) | 0.443 |
| Adult **always** washes hands with soap before feeding child (%) | 233 (64.2) | 247 (68.2) | 0.25 |
| **Always** use hand sanitiser for child when dining outside (%) | 127 (35.0) | 109 (30.1) | 0.161 |
| **Always** use sanitiser for baby seats when dining outside (%) | 116 (32.0) | 113 (31.2) | 0.83 |
| Wash toys with soap at home (%) | | | **0.004** |
| Never | **94 (25.9)** | **107 (29.6)** | |
| Once every 2 to 3 weeks | **203 (55.9)** | **173 (47.8)** | |
| Once a week | **52 (14.3)** | **46 (12.7)** | |
| More than once a week | **14 (3.9)** | **36 (9.9)** | |
| Use sanitiser to clean toys at home (%) | | | **0.045** |
| Never | **152 (41.9)** | **162 (44.8)** | |
| Once every 2 to 3 weeks | **147 (40.5)** | **117 (32.3)** | |
| Once a week | **47 (12.9)** | **52 (14.4)** | |
| More than once a week | **17 (4.7)** | **31 (8.5)** | |
| Household cleaning frequency (%) | | | 0.052 |
| Less than once a week | 24 (6.6) | 30 (8.3) | |
| Once a week | 141 (38.9) | 109 (30.1) | |
| More than once a week | 61 (16.9) | 81 (22.4) | |
| Every day | 136 (37.6) | 142 (39.2) | |
| Share utensils with siblings at home (%) | **184 (68.4)** | **142 (59.7)** | **0.040** |
| Knowledge of HFMD (%) | | | |
| Question 1: Transmission route | 351 (96.7) | 340 (93.9) | 0.078 |
| Question 2: Signs and symptoms | **362 (99.7)** | **355 (98.1)** | **0.033** |
| Question 3: Incubation period | **343 (94.5)** | **323 (89.2)** | **0.010** |
| Question 4: Susceptibility (adult) | **357 (98.4)** | **338 (93.4)** | **0.001** |
| Question 5: Susceptibility (children) | 347 (95.6) | 335 (92.5) | 0.082 |
| Question 6: Isolation | 362 (99.7) | 361 (99.7) | 0.998 |
| Question 7: Hygiene | 362 (99.7) | 361 (99.7) | 0.998 |

*(Continued)*

**Table 3.** (Continued)

| | Cases | Controls | *p*-value* |
|---|---|---|---|
| | **(n = 363)** | **(n = 362)** | |
| Question 8: Treatment | **346 (95.3)** | **309 (85.4)** | **< 0.001** |
| Question 9: Vaccine | **326 (89.8)** | **288 (79.6)** | **< 0.001** |
| Question 10: Inform childcare centre | 362 (99.7) | 362 (100.0) | 0.318 |
| Question 11: Return to childcare centre | 346 (95.3) | 340 (93.9) | 0.405 |
| Question 12: Sharing articles | 361 (99.5) | 359 (99.2) | 0.651 |
| Question 13: Disinfection | 357 (98.4) | 350 (96.7) | 0.150 |
| Question 14: Virus shedding in stool | 160 (44.1) | 165 (45.6) | 0.684 |
| Question 15: Virus shedding in saliva | 167 (46.0) | 173 (47.8) | 0.630 |
| Knowledge score of HFMD (mean ± SD) | **13.5 ± 1.4** | **13.1 ± 1.8** | **0.002** |

Values are means ± SD or n (%).

* Student's t-tests or chi-square tests.

a Account for children age > 1.5 years old.

HFMD, hand, foot and mouth disease; HA, herpangina; SD, standard deviation.

multivariate analysis as these effects were not significant after controlled for other confounding factors so as to achieve a more parsimonious model (Table 5).

The crude OR of HFMD was 1.19 per year increase in age (95% CI 1.07 to 1.31, *p* = 0.001). Age group was assessed for *p*-trend and a significant linear trend was found (*p* < 0.001). Age 3–3.9 years had the highest risk of HFMD (OR = 5.27, 95% CI 2.47 to 11.22, *p* < 0.001), followed by 4–4.9 years (OR = 3.44, 95% CI 1.53 to 7.77, *p* = 0.003) and 1.5–2.9 years (OR = 2.97, 95% CI 1.44 to 6.14, *p* = 0.003). However, adjusted ORs for age 5–5.9 years and 6–6.9 years were attenuated and effects became not significant. The adjusted OR of HFMD was 1.53 per year increase in preschool admission period (95% CI 1.25 to 1.86, *p* < 0.001). The effects of gender, nationality and ethnicity were not significant in both univariate and multivariate analysis.

Gestational age of 37–42 weeks had an increased risk of HFMD (OR = 1.79, 95% CI 1.00 to 3.20, *p* = 0.049) compared to < 37 weeks after adjusted for confounders. No significant effect on HFMD was found for birth weight, mode of child delivery, mother's age at child birth, and breastfeeding. Compared to starting solids at 4–6 months old, starting solids at 10 months old and above (OR = 0.52, 95% CI 0.30 to 0.90, *p* = 0.021) showed decreasing risk of HFMD after adjusted for confounders.

Univariate analysis showed that having one sibling (crude OR = 1.76, 95% CI 1.28 to 2.41, *p* < 0.001) and 2–3 siblings (crude OR = 1.95, 95% CI 1.01 to 3.76, *p* = 0.045) in the same childcare centre had increasing risk of HFMDcompared to no sibling in the same childcare centre. However, both effects were attenuated to below the null and no longer significant after controlled for confounders. In both univariate and multivariate model, having HFMD-infected siblings had higher risk of HFMD (crude OR = 3.37, 95% CI 2.44 to 4.66, *p* <0.001; adjusted OR = 3.51, 95% CI 2.47 to 5.00, *p* < 0.001, respectively) than no siblings with history of HFMD. Child with history of chicken pox was associated with 2-fold higher risk of HFMD (crude OR = 2.07, 95% CI 1.09 to 3.92, *p* = 0.026) than child who never had chicken pox but the association was not significant in the multivariate analysis. Child with rotavirus vaccination had higher risk of HFMD than child without rotavirus vaccination (crude OR = 1.46, 95% CI 1.06 to 2.00, *p* = 0.021) but the association was not significant after adjustment for potential confounding factors.

**Table 4. Characteristics of childcare centre.**

|  | Cases | Controls | p-value* |
|---|---|---|---|
|  | (n = 363) | (n = 362) |  |
| Region (%) |  |  | 0.057 |
| North | 153 (42.2) | 147 (40.6) |  |
| South | 47 (13.0) | 32 (8.8) |  |
| East | 92 (25.3) | 123 (34.0) |  |
| West | 52 (14.3) | 48 (13.3) |  |
| Central | 19 (5.2) | 12 (3.3) |  |
| Building type (%) |  |  | 0.673 |
| Residential | 212 (58.4) | 217 (59.9) |  |
| Commercial | 151 (41.6) | 145 (40.1) |  |
| Classroom layout (%) |  |  | 0.667 |
| Enclosed | 109 (30.0) | 107 (29.5) |  |
| Open-concept | 87 (24.0) | 97 (26.8) |  |
| Mixed enclosed & open-concept | 167 (46.0) | 158 (43.7) |  |
| Playground (%) |  |  | 0.501 |
| None | 59 (16.2) | 51 (14.1) |  |
| Indoor | 62 (17.1) | 53 (14.6) |  |
| Outdoor | 226 (62.3) | 245 (67.7) |  |
| Both | 16 (4.4) | 13 (3.6) |  |
| Cleaner (%) |  |  | **0.001** |
| In-house | **20 (5.5)** | **45 (12.4)** |  |
| Out-source | **343 (94.5)** | **317 (87.6)** |  |
| Ventilation (%) |  |  | 0.687 |
| Air-conditioned | 27 (7.4) | 28 (7.7) |  |
| Fan | 127 (35.0) | 137 (37.9) |  |
| Both | 209 (57.6) | 197 (54.4) |  |
| Enrolment (%) |  |  | **0.038** |
| 51–86 | **106 (29.2)** | **129 (35.6)** |  |
| 87–115 | **151 (41.6)** | **155 (42.8)** |  |
| 116–260 | **106 (29.2)** | **78 (21.6)** |  |
| Class size (%) |  |  | 0.086 |
| 11–15 | 99 (27.3) | 120 (33.2) |  |
| 16–21 | 152 (41.9) | 154 (42.5) |  |
| 22–44 | 112 (30.8) | 88 (24.3) |  |
| Classroom with hand sanitiser (%) | 185 (51.0) | 205 (56.6) | 0.126 |

Values are n (%).

* Chi-square tests.

Child taken care by grandparents when sick was at higher risk of HFMD (crude OR = 1.64, 95% CI 1.20 to 2.24, $p$ = 0.002) compared to child not taken care by grandparents when sick but the adjusted OR was weakened and not significant. Mother's age at 41–50 years old (OR = 0.30, 95% CI 0.15 to 0.60, $p$ = 0.001) showed decreasing risk of HFMD compared to age 21–30 years in the multivariate model. Univariate analysis showed that having two children in the family had higher risk of HFMD compared to only one child (crude OR = 1.50, 95% CI 1.07 to 2.09, $p$ = 0.018). However after controlling for confounders, having 2 or >2 children in the family had lower risk of HFMD (OR = 0.65, 95% CI 0.43 to 0.99, $p$ = 0.045; OR = 0.52, 95%

**Table 5. Univariate and multivariate adjusted regression models.**

| | Unadjusted | *p*-value | Adjusted [*] | *p*-value |
|---|---|---|---|---|
| | OR (95% CI) | | OR (95% CI) | |
| Age at recruitment (years) | **1.19 (1.07, 1.31)** | **0.001** | | |
| Age group at recruitment | | **0.005[a]** | | |
| 0.4–1.5 years | **1.00** | | **1.00** | |
| 1.5–2.9 years | **3.38 (1.69, 6.77)** | **0.001** | **2.97 (1.44, 6.14)** | **0.003** |
| 3–3.9 years | **6.43 (3.19, 12 96)** | **<0.001** | **5.27 (2.47, 11.22)** | **< 0.001** |
| 4–4.9 years | **6.21 (3.05, 12.65)** | **<0.001** | **3.44 (1.53, 7.77)** | **0.003** |
| 5–5.9 years | **3.90 (1.89, 8.04)** | **<0.001** | 1.77 (0.75, 4.19) | 0.195 |
| 6–6.9 years | **3.37 (1.37, 8.28)** | **0.008** | 1.15 (0.39, 3.42) | 0.797 |
| Preschool admission period (years) | **1.45 (1.26, 1.67)** | **< 0.001** | **1.53 (1.25, 1.86)** | **< 0.001** |
| Infected once | **1.26 (1.07, 1.47)** | **0.004** | **1.46 (1.15, 1.84)** | **0.002** |
| Infected more than once | **1.64 (1.38, 1.95)** | **< 0.001** | **1.54 (1.20, 1.97)** | **0.001** |
| Gender | | | | |
| Male | 1.00 | | 1.00 | |
| Female | 0.81 (0.60, 1.08) | 0.148 | 0.75 (0.54, 1.05) | 0.093 |
| Nationality | | | | |
| Singaporean | 1.00 | | 1.00 | |
| Non-Singaporean | 0.48 (0.23, 1.01) | 0.054 | 0.79 (0.36, 1.73) | 0.554 |
| Ethnicity | | | | |
| Chinese | 1.00 | | 1.00 | |
| Malay | 0.80 (0.52, 1.24) | 0.321 | 0.92 (0.56, 1.51) | 0.748 |
| Indian | 0.54 (0.21, 1.39) | 0.205 | 0.46 (0.17, 1.28) | 0.139 |
| Others | 0.58 (0.28, 1.17) | 0.128 | 0.88 (0.40, 1.96) | 0.764 |
| Birth weight | | | | |
| < 2.5 kg | 1.00 | | 1.00 | |
| 2.5–3.4 kg | 0.97 (0.57, 1.65) | 0.909 | 1.13 (0.62, 2.06) | 0.678 |
| > 3.4 kg | 1.20 (0.66, 2.19) | 0.554 | 1.64 (0.83, 3.24) | 0.155 |
| Gestational age | | | | |
| < 37 weeks | 1.00 | | 1.00 | |
| 37–42 weeks | 1.35 (0.80, 2.25) | 0.258 | **1.79 (1.00, 3.20)** | **0.049** |
| Mode of delivery | | | | |
| Natural birth | 1.00 | | 1.00 | |
| Caesarean section | 0.98 (0.72, 1.32) | 0.878 | 1.01 (0.72, 1.41) | 0.962 |
| Mother's age at child birth | | | | |
| < 29 years old | 1.00 | | 1.00 | |
| 30–34 years old | 0.86 (0.62, 1.20) | 0.382 | 0.80 (0.50, 1.28) | 0.364 |
| > 34 years old | 0.69 (0.46, 1.03) | 0.070 | 1.03 (0.56, 1.89) | 0.926 |
| Breastfeeding | | | | |
| Never breastfeed | 1.00 | | 1.00 | |
| Mixed breastfeed with infant formula | 1.55 (0.82, 2.92) | 0.173 | 1.58 (0.77, 3.24) | 0.210 |
| Exclusive breastfeed for 2 months or more | 1.53 (0.37, 0.98) | 0.191 | 1.46 (0.71, 3.01) | 0.300 |
| Age of starting solids | | **0.034[a]** | | |
| 4–6 months old | 1.00 | | 1.00 | |
| 7–9 months old | 0.81 (0.59, 1.11) | 0.182 | 0.74 (0.51, 1.05) | 0.094 |
| Above 10 months old | **0.61 (0.37, 0.98)** | **0.042** | **0.52 (0.30, 0.90)** | **0.021** |
| Siblings in same childcare centre | | **<0.001[a]** | | |
| 0 | 1.00 | | 1.00 | |

(*Continued*)

**Table 5.** (*Continued*)

| | Unadjusted | *p*-value | Adjusted * | *p*-value |
|---|---|---|---|---|
| | OR (95% CI) | | OR (95% CI) | |
| 1 | **1.76 (1.28, 2.41)** | **< 0.001** | 0.95 (0.64, 1.42) | 0.805 |
| 2–3 | **1.95 (1.01, 3.76)** | **0.045** | 0.80 (0.36, 1.78) | 0.592 |
| Siblings HFMD infection history (before the identified | | | | |
| No | 1.00 | | 1.00 | |
| Yes | **3.37 (2.44, 4.66)** | **< 0.001** | **3.51 (2.47, 5.00)** | **< 0.001** |
| Flu in the last 3 months | 1.09 (0.81, 1.49) | 0.566 | 1.23 (0.86, 1.76) | 0.244 |
| Diarrheal in the last 3 months | 0.93 (0.61, 1.42) | 0.734 | 1.13 (0.70, 1.82) | 0.615 |
| History of chicken pox | **2.07 (1.09, 3.92)** | **0.026** | 1.49 (0.74, 2.97) | 0.259 |
| History of eczema | 1.19 (0.71, 2.01) | 0.508 | 1.54 (0.85, 2.79) | 0.150 |
| Optional rotavirus vaccination | **1.46 (1.06, 2.00)** | **0.021** | 1.37 (0.96, 1.96) | 0.078 |
| Main caretaker when sick | | | | |
| Parents | 0.92 (0.63, 1.33) | 0.65 | 1.11 (0.74, 1.68) | 0.608 |
| Grandparents | **1.64 (1.20, 2.24)** | **0.002** | 1.40 (0.99, 1.98) | 0.058 |
| Domestic helper | 1.04 (0.71, 1.52) | 0.858 | 0.86 (0.56, 1.33) | 0.501 |
| Mother's age | | | | |
| 21–30 years | 1.00 | | 1.00 | |
| 31–40 years | 1.11 (0.75, 1.64) | 0.608 | 0.91 (0.59, 1.41) | 0.688 |
| 41–50 years | 0.54 (0.29, 1.01) | 0.053 | **0.30 (0.15, 0.60)** | **0.001** |
| Father's age | | | | |
| 21–30 years | 1.00 | | 1.00 | |
| 31–40 years | 1.14 (0.63, 2.05) | 0.665 | 0.88 (0.41, 1.91) | 0.753 |
| 41–60 years | 0.80 (0.42, 1.51) | 0.494 | 0.60 (0.25, 1.44) | 0.257 |
| Total children | | | | |
| 1 | 1.00 | | 1.00 | |
| 2 | **1.50 (1.07, 2.09)** | **0.018** | **0.65 (0.43, 0.99)** | **0.045** |
| > 2 | 1.47 (0.95, 2.29) | 0.084 | **0.52 (0.30, 0.93)** | **0.027** |
| Household size | | | | |
| 2 | 1.00 | | 1.00 | |
| 3 | 0.96 (0.15, 5.89) | 0.963 | 1.34 (0.17, 10.82) | 0.783 |
| 4 | 1.79 (0.29, 10.87) | 0.528 | 1.31 (0.17, 10.40) | 0.796 |
| > 4 | 1.63 (0.27, 9.89) | 0.596 | 1.03 (0.13, 8.18) | 0.974 |
| Household members | | | | |
| Parents and child | 1.00 | | 1.00 | |
| Parents, child and domestic helper | 1.21 (0.84, 1.74) | 0.302 | 0.95 (0.62, 1.43) | 0.793 |
| Grandparents, parents and child | **1.55 (1.01, 2.38)** | **0.045** | 1.41 (0.88, 2.27) | 0.152 |
| Grandparents, parents, child and domestic helper | 0.59 (0.32, 1.07) | 0.084 | **0.50 (0.26, 0.96)** | **0.039** |
| Highest education of parents | | | | |
| Non-tertiary | 1.00 | | 1.00 | |
| Tertiary & above | **1.80 (1.16, 2.80)** | **0.009** | **1.95 (1.19, 3.18)** | **0.008** |
| Housing type | | | | |
| Public housing | 1.00 | | 1.00 | |
| Private housing | 0.94 (0.65, 1.35) | 0.723 | 0.87 (0.58, 1.32) | 0.523 |
| Housing space | | | | |
| 1 to 3 rooms | 1.00 | | 1.00 | |
| > 3 rooms | 0.75 (0.53, 1.06) | 0.106 | 0.71 (0.48, 1.06) | 0.095 |
| Household monthly income | | | | |

(*Continued*)

**Table 5.** (*Continued*)

| | Unadjusted | p-value | Adjusted * | p-value |
|---|---|---|---|---|
| | OR (95% CI) | | OR (95% CI) | |
| < S$8,000 | 1.00 | | 1.00 | |
| S$8,000 –$10,000 | 1.11 (0.73, 1.66) | 0.631 | 1.04 (0.65, 1.66) | 0.862 |
| > S$10,000 | 0.85 (0.61, 1.20) | 0.361 | 0.75 (0.50, 1.12) | 0.167 |
| Parents with diabetes | | | | |
| Mother or father | 1.35 (0.63, 2.89) | 0.442 | 1.17 (0.49, 2.78) | 0.728 |
| Both | 2.02 (0.18, 22.41) | 0.566 | 0.54 (0.05, 6.31) | 0.628 |
| Parents with hypertension | | | | |
| Mother or father | 0.77 (0.41, 1.46) | 0.428 | 0.78 (0.37, 1.64) | 0.517 |
| Both | 2.96 (0.31, 28.64) | 0.348 | 4.65 (0.28, 87.34) | 0.305 |
| Parents with hyperlipidaemia | | | | |
| Mother or father | 0.61 (0.25, 1.49) | 0.275 | 0.60 (0.21, 1.74) | 0.349 |
| Both | 2.96 (0.31, 28.65) | 0.347 | 2.93 (0.26, 32.33) | 0.38 |
| Parents with HFMD previously^ | | | | |
| Mother or father | **3.89 (2.30, 6.57)** | **< 0.001** | **3.11 (1.75, 5.53)** | **< 0.001** |
| Both | 2.94 (0.91, 9.49) | 0.070 | **4.05 (1.07, 15.23)** | **0.039** |
| Plays in outdoor playground | | 0.847[a] | | |
| Never | 1.00 | | 1.00 | |
| Once every 2 to 3 weeks | **2.66 (1.53, 4.61)** | **0.001** | 1.45 (0.75, 2.80) | 0.264 |
| Once a week | **1.79 (1.02, 3.13)** | **0.041** | 1.12 (0.57, 2.20) | 0.744 |
| More than once a week | **1.84 (1.06, 3.18)** | **0.029** | 1.03 (0.53, 2.00) | 0.939 |
| Plays in indoor playground | | 0.486[a] | | |
| Never | 1.00 | | 1.00 | |
| Once every 2 to 3 weeks | 1.43 (0.98, 2.08) | 0.06 | 1.04 (0.67, 1.61) | 0.867 |
| Once a week | 0.93 (0.56, 1.54) | 0.768 | 0.68 (0.38, 1.23) | 0.203 |
| More than once a week | 0.90 (0.49, 1.68) | 0.75 | 0.67 (0.33, 1.34) | 0.257 |
| Plays with other children in the neighbourhood | | 0.994[a] | | |
| Never | 1.00 | | 1.00 | |
| Once every 2 to 3 weeks | 1.09 (0.75, 1.61) | 0.63 | 0.83 (0.54, 1.29) | 0.416 |
| Once a week | 0.79 (0.52, 1.22) | 0.292 | 0.72 (0.44, 1.16) | 0.18 |
| More than once a week | 1.09 (0.74, 1.61) | 0.662 | 0.83 (0.53, 1.30) | 0.42 |
| Child **always** washes hands with soap before eating | 0.78 (0.58, 1.07) | 0.123 | 0.91 (0.65, 1.28) | 0.58 |
| Child **always** washes hands with soap after eating | 0.96 (0.70, 1.31) | 0.805 | 1.08 (0.78, 1.53) | 0.646 |
| Child **always** washes hands with soap after toilet | 1.12 (0.82, 1.54) | 0.476 | 1.24 (0.87, 1.76) | 0.227 |
| Adult **always** washes hands with soap after changing diaper or washing up child after toilet | 0.86 (0.58, 1.26) | 0.443 | 0.93 (0.61, 1.42) | 0.727 |
| Adult **always** washes hands with soap before feeding child | 0.83 (0.61, 1.14) | 0.25 | 1.00 (0.71, 1.41) | 0.983 |
| **Always** use hand sanitiser for child when dining outside | 1.25 (0.91, 1.70) | 0.162 | 1.34 (0.94, 1.89) | 0.104 |
| **Always** use sanitiser for baby seats when dining outside | 1.03 (0.76, 1.41) | 0.83 | 1.36 (0.95, 1.95) | 0.092 |
| Wash toys with soap at home | | 0.263[a] | | |
| Never | 1.00 | | 1.00 | |
| Once every 2 to 3 weeks | 1.34 (0.95, 1.88) | 0.098 | **1.51 (1.03, 2.22)** | **0.035** |
| Once a week | 1.29 (0.79, 2.09) | 0.307 | 1.40 (0.81, 2.42) | 0.231 |
| More than once a week | **0.44 (0.22, 0.87)** | **0.018** | 0.70 (0.34, 1.46) | 0.347 |
| Use sanitiser to clean toys at home | | 0.346[a] | | |
| Never | 1.00 | | 1.00 | |
| Once every 2 to 3 weeks | 1.34 (0.96, 1.86) | 0.082 | **1.63 (1.12, 2.36)** | **0.01** |
| Once a week | 0.96 (0.61, 1.51) | 0.871 | 1.10 (0.66, 1.82) | 0.711 |

(*Continued*)

**Table 5.** (Continued)

| | Unadjusted | p-value | Adjusted * | p-value |
|---|---|---|---|---|
| | OR (95% CI) | | OR (95% CI) | |
| More than once a week | 0.58 (0.31, 1.10) | 0.096 | 0.99 (0.49, 1.99) | 0.977 |
| Household cleaning frequency | | 0.338[a] | | |
| Less than once a week | 1.00 | | 1.00 | |
| Once a week | 1.62 (0.89, 2.92) | 0.112 | 1.62 (0.83, 3.15) | 0.153 |
| More than once a week | 0.94 (0.50, 1.77) | 0.851 | 1.11 (0.55, 2.25) | 0.774 |
| Everyday | 1.20 (0.67, 2.15) | 0.547 | 1.06 (0.55, 2.04) | 0.866 |
| Share utensils with siblings at home | **1.46 (1.02, 2.12)** | **0.041** | 1.13 (0.74, 1.71) | 0.569 |
| Knowledge score of HFMD | **1.16 (1.05, 1.27)** | **0.002** | **1.11 (1.00, 1.24)** | **0.047** |
| Childcare centre building type | | | | |
| Residential | 1.00 | | 1.00 | |
| Commercial | 1.06 (0.79, 1.43) | 0.673 | 1.02 (0.72, 1.44) | 0.924 |
| Classroom layout | | | | |
| Enclosed | 1.00 | | 1.00 | |
| Open concept | 0.88 (0.59, 1.30) | 0.526 | 1.07 (0.68, 1.67) | 0.774 |
| Both | 1.04 (0.73, 1.46) | 0.834 | 1.12 (0.76, 1.64) | 0.573 |
| Playground in childcare centre | | | | |
| None | 1.00 | | 1.00 | |
| Indoor | 1.01 (0.60, 1.71) | 0.967 | 1.02 (0.57, 1.84) | 0.945 |
| Outdoor | 0.80 (0.52, 1.21) | 0.286 | 0.72 (0.45, 1.17) | 0.186 |
| Both | 1.06 (0.47, 2.42) | 0.883 | 0.83 (0.33, 2.11) | 0.701 |
| Cleaner in childcare centre | | | | |
| In-house | 1.00 | | 1.00 | |
| Out-source | **2.43 (1.40, 4.21)** | **0.001** | **2.55 (1.39, 4.68)** | **0.002** |
| Ventilation in childcare centre | | | | |
| Air-conditioned | 1.00 | | 1.00 | |
| Fan | 0.96 (0.54, 1.72) | 0.894 | 0.78 (0.40, 1.51) | 0.462 |
| Both | 1.10 (0.63, 1.93) | 0.74 | 0.91 (0.48, 1.72) | 0.772 |
| Enrolment of childcare centre | | 0.012[a] | | |
| 51–86 | 1.00 | | 1.00 | |
| 87–115 | 1.18 (0.84, 1.67) | 0.328 | 1.39 (0.94, 2.05) | 0.097 |
| 116–260 | **1.65 (1.12, 2.44)** | **0.011** | **1.74 (1.13, 2.68)** | **0.012** |
| Classroom size of childcare centre | | 0.028[a] | | |
| 11–15 | 1.00 | | 1.00 | |
| 16–21 | 1.20 (0.84, 1.69) | 0.312 | 1.36 (0.92, 2.01) | 0.128 |
| 22–44 | **1.54 (1.05, 2.27)** | **0.028** | **1.59 (1.03, 2.45)** | **0.035** |
| Classroom with hand sanitiser | 0.79 (0.59, 1.07) | 0.126 | 0.92 (0.66, 1.28) | 0.633 |

Odds ratios derived from unconditional logistic regression models.

[a] p-trend test.

* Model adjusted for age group, education, mother's age, HFMD-infected siblings, and preschool admission period.

^ At the point the questionnaire was performed.

OR, odds ratio; CI, confidence intervals; HFMD, hand, foot and mouth disease.

CI 0.30 to 0.93, $p = 0.027$) compared to only 1 child. ORs for household size, housing space, and household monthly income were not significant in both univariate and multivariate analysis.

Univariate analysis showed that household members consist of grandparents, parents and child had increased risk of HFMD (crude OR = 1.55, 95% CI 1.01 to 2.38, p = 0.045) compared to household members consist of parents and child only. Multivariate analysis showed that household members consist of grandparents, domestic helper, parents and child showed a lower risk of HFMD (OR = 0.50, 95% CI 0.26 to 0.96, $p$ = 0.039). The risk of HFMD in children whose parents had tertiary education and above were 1.95 times higher than children whose parents had non-tertiary education (OR = 1.95, 95% CI 1.19 to 3.18, $p$ = 0.008) after adjusted for confounders. In the multivariate analysis, oneparent and both parents who had HFMD previously showed higher risk of HFMD(OR = 3.11, 95% CI 1.75 to 5.53, $p$< 0.001; OR = 4.05, 95% CI 1.07 to 15.23, $p$ = 0.039, respectively) compared to parents who did not ever had HFMD.

In the univariate model, children who play in outdoor playground had higher risk of HFMD compared to children who never play in outdoor playground, with decreasing risk when frequency of play increased. However, association of play in outdoor playground and HFMD was not significant in the multivariate model. Both univariate and multivariate analysis showed no significant effects for child plays in indoor playground; child plays with neighbour-hood children; child's frequent hand washing with soap; adult's frequent hand washing with soap; frequent use of hand sanitiser for child; and frequent use of sanitiser for baby seats. Comparedto never wash toys with soap at home, wash toys with soap more than once a week was protective in the univariate model (crude OR = 0.44, 95% CI 0.22 to 0.87, $p$ = 0.018) but not significant after adjusted. However, multivariate analysis showed that wash toys with soap once every 2 to 3 weeks had higher risk of HFMD (OR = 1.51, 95% CI 1.03 to 2.22, $p$ = 0.035) compared to never wash toys with soap. Compared to never use sanitiser to clean toys, sanitise toys once every 2 to 3 weeks had higher risk of HFMD (OR = 1.63, 95% CI 1.12 to 2.36, $p$ = 0.010) after controlled for confounders. No significant association was found for household cleaning frequency and HFMD. In the univariate model, sharing utensils with siblings at home had 1.46 times higher risk of HFMD (95% CI 1.02 to 2.12, $p$ = 0.041) compared to never share utensils with siblings but the effect was not significant after adjusted. The crude OR of HFMD was 1.16 per score increase in knowledge of HFMD (95% CI 1.05 to 1.27, $p$ = 0.002) but effect was weakened (OR = 1.11, 95% CI 1.00 to 1.24, $p$ = 0.047) in the multivariate analysis.

Building type, classroom layout, playground type, classroom with hand sanitiser and ventilation of childcare centre had no significant effects on HFMD. Children in childcare centre with out-sourced cleaners had significantly higher risk of HFMD compared to children in childcare centre with in-house cleaners (OR = 2.55, 95% CI 1.39 to 4.68, $p$ = 0.002) after adjusted for confounders. Children in childcare centre with 116–260 children had increased risk of HFMD (OR = 1.74, 95% CI 1.13 to 2.68, $p$ = 0.012) compared to children in childcare centre with 51–86 children in the multivariate model. A significant risk increased was found for class size of 22–44 children (OR = 1.59, 95% CI 1.03 to 2.45, $p$ = 0.035) compared to class size of 11–15 children after adjustment for potential confounders.

## Discussion

### Age distribution of HFMD

This study showed that child aged 3 to 3.9 years had the highest risk of HFMD infections which corresponded to the age group with the highest incidence rate in Singapore [6, 15, 23]. Children have different immunity status across different age groups. Infants below one year old may have protection from passive immunity of immuno globulins (Ig) G and IgA antibodies against bacterial and viral infections [44]. The increased herd immunity among older

children aged 5 years and above might have been due to their previous infection with HFMD thus protecting them and peers against HFMD infections [6, 45].

## Gender and ethnic distribution of HFMD

There was no statistically significant gender-specific differences of HFMD in this study as supported byserologic evidence based on a local seroprevalence study in Singapore [30]. The overall male to female ratio in this study was 1.2: 1 which aligned with the recent national disease surveillance statistics of HFMD [15]. Nonetheless, male cases were predominant during 2001 to 2007 HFMD outbreaks in Singapore [20, 21]. Taken together, HFMD infections of both male and female is comparable in Singapore. No significant risk effect of nationality and ethnicity was found in this study. However, national disease surveillance statistics showed that Malay ethnic group had the highest incidence rate of HFMD, followed by Chinese and Indians [15,21]. A study by Ang et al. also found that Malays had the highest seroprevalence of HFMD enteroviruses among the three major ethnic groups in Singapore [2]. This study did not show Malay as a risk factor possibly due to the lower proportion of Malay ethnicity in these childcare centres.

## Maternal, birth and infancy factors with HFMD

The association of mother's gestational age and HFMD infections was not significant in this study. A study by Zhu et al. reported that longer gestational age of 37 to 42 weeks was protective against incidence of fever in children with HFMD [39]. Although shorter gestational age had been linked to poor innate immunity in preterm infants [46], evidence in this study suggest that the protective effect from maternal antibodies among infants with gestational age of 37–42 weeks may not be higher than infants with gestational age of less than 37 weeks in terms of HFMD infections. No evidence of exclusive breastfeeding protects against of HFMD in this study. Prospective cohort studies had shown protective effect of breastfeeding against enterovirus infections during infancy period [44, 47, 48]. Lin et al. found that exclusive breastfeeding was a protective factor (OR = 0.63, 95% CI 0.47 to 0.85, $p$ = 0.002) of HFMD infections compared to mixed breastfeeding but the protection only last for age 6 to 28 months [37]. Protection from breastfeeding depends on the volume of breast milk ingested and the duration of breastfeeding but weakens over age due to decreasing concentration of immune components in breast milk over time [48]. Compared to China with about 13.6% exclusive breastfeeding for infants at 6 months old according to a cross-sectional survey in 2010 [49], Singapore had only 1% exclusively breastfed infants at 6 months of age as reported from the National Breastfeeding Survey in 2011 [50]. Hence, the effect of exclusive breastfeeding may not be significantly shown based on local context with a very low exclusive breastfeeding rate.

Starting solids at 10 months old and above during infancy period was likely to be more protective of HFMD infections than starting solids earlier at 4 to 6 months old. This finding may indirectly relates weaning age to the diet and nutrition including longer breastfeeding duration which may have higher effect to build immunity of infants. Nonetheless, no studies had investigated the association of weaning age and HFMD infections. Infants are gradually introduced with solid foods other than breast milk and formula milk to supplement daily nutrition requirements. World Health Organization (WHO) recommended the introduction of complementary foods to infants should be after the first 6 months of exclusive breastfeeding [51]. However, based on a systematic review, there was no clear evidence to show the association of weaning age and infections among infants [52]. Another possibility may be due to the increase risk of cross-contamination from environment during food preparation, hence, increasing the

risk of HFMD when starting solid early as compared to later at 10 months old where immunity of the baby is also stronger.

## Household contacts and HFMD

Having siblings with HFMD infections increased the risk of HFMD infections due to increased exposure and high transmission rate among close contacts at home. A local study suggested that the risk of HFMD transmission within the same household was the greatest during HFMD outbreak [19]. During an epidemic in Taiwan, close contact within a family was more likely to develop HFMD than non-family contacts [29]. In this study, parents with history of HFMD infection were also strongly associated with HFMD infections in children. Hence, genetic susceptibility may be involved in the risk of HFMD infection. Chang et al. found that more children in a family increased the risk of HFMD infections by 40% [29]. However, having more children in a family was not a risk factor that attribute HFMD transmission in this study. This may suggest that the impact of total children in a family on HFMD infections is lesser compared to having HFMD-infected siblings, as there may be already some children who were infected before and hence, reduced the risk of infecting another child in the family. HFMD transmission is more likely when a child in the family is infected with HFMD, regardless of the number of children in the family. Likewise, household size was not a risk factor of HFMD in this study. Having grandparents, parents, child and domestic helper in the same household had significantly lower risk of HFMD infections as compared to only parents and child in a household. The presence of domestic helper may suggest more thorough and frequent household cleaning thus minimizes the exposure to HFMD viruses. Other household factors, housing environment, and parents' socioeconomic status had no significant effect on risk of HFMD infections.

## Playground, neighbourhood exposure and HFMD

No significant associations was found between child who played in outdoor/indoor playground or played with neighbourhood children and HFMD. Increased frequency of both factors did not increased the risk of HFMD infections suggest that exposure to public playgrounds may not be a risk factor in Singapore. This finding was not consistent with a matched case-control study which found that child who played with neighbourhood children had 11 times higher risk of HFMD compared to never play with neighbourhood children [33]. Another case-control study in China reported similar adjusted ORs for children who went to outdoor and indoor playground with 2.3 times higher risk of HFMD [53]. Xie et al. reported that EV-A71 can be inactivated by heat exposure of 60°C and ultraviolet (UV) irradiation [54]. Hence, there may be adequate disinfection as outdoor playground exposed to UV irradiation from the sun all year round in Singapore.

## Hygiene practices and HFMD

No significant HFMD risk reduction was found with frequent hand washing with soap for both child and adult in this study. Many studies had reported that hand washing is effective to reduce HFMD infections in children. Sun et al. reported that children's hand washing habits (before dinner and after using toilet) and adult's hand washing before in contact with children were protective factors of HFMD [53]. Ruan et al. found that almost always hand washing before meal among children and before feeding child among adults reduced more than 95% risk of HFMD with significantly increased dose-response effect after adjusting for other factors [33]. Zhang et al. reported that often hand washing before meals reduced HFMD risk significantly but not hand washing after toilet use [55]. Nevertheless, HFMD continued to spread

even though hand hygiene education was strongly emphasised. HFMD viruses may still persist on contaminated hands and surfaces for prolonged period of time if hand washing is not effective due to many reasons. This may suggest that hand washing is necessary but may not be sufficient for effective hand washing to prevent HFMD transmission during an outbreak. Studies showed that the positive rate of enteroviruses in throat swabs was higher than rectal swabs during an outbreak [56, 57] which suggest that HFMD transmission may not necessary be via faecal-oral route during acute stage of HFMD when saliva excretion or respiratory droplets are highly contagious. In addition, improving hand washing technique by following proper hand washing steps is as important as hand hygiene education. Centers for Disease Control and Prevention (CDC) recommended hand hygiene technique in health-care settings: "When washing hands with soap and water, wet hands first with water, apply an amount of product recommended by the manufacturer to hands, and rub hands together vigorously for at least 15 seconds, covering all surfaces of the hands and fingers. Rinse hands with water and dry thoroughly with a disposable towel. Use towel to turn off the faucet" [58].

This study did not find the protective effect of frequent use of hand sanitiser against HFMD infections suggest that hand hygiene using hand sanitisers may not be effective for HFMD prevention. Chang et al. found that 95% ethanol exhibited the highest virucidal effect against EV-A71 but not complete inactivation [59]. However, most alcohol-based hand sanitisers in the market that contain up to 70% ethanol may not be effective for total inactivation of HFMD viruses. Alcohol-free hand sanitisers may contain antiseptic agents such as chlorhexidine gluconate or benzalkonium chloride. A study reported that benzalkonium chloride (quaternary ammonium compound) in a Dettol Hospital Disinfectant inactivated human coxsackie virus with a reduction of $>5 \log_{10}$ after one minute contact time, subjected to the concentration of active agent and pH [60]. Nevertheless, only limited studies using laboratory suspension testing method instead of randomized controlled trial are available to show the effectiveness of active agent in hand sanitiser against human enteroviruses which causes HFMD.

Frequent wash toys with soap, frequent use of sanitiser to clean toys, and frequent household cleaning were not protective of HFMD infections suggest that the frequency may not be sufficient for effective cleaning and sanitising. Ineffective way to clean and sanitise toys may increase the risk of HFMD transmission as other toys will be cross-contaminated. Enteroviruses are non-enveloped viruses with prolonged environmental survival and able to resist common disinfectants such as organic solvents, 1% quaternary ammonium compound, 5% phenols, and 70% ethanol [61, 62]. A study reported that 3,120 parts per million (ppm) of sodium hypochlorite (0.312%) with 5 minutes contact time inactivated both EV-A71 and CV-A16 effectively [63]. Household bleach usually contains 5–6% sodium hypochlorite. Using diluted household bleach (1:10 dilution or 1 part + 9 parts of water) to disinfect contaminated surfaces/articles was recommended by the MOH in the Infection Control Guidelines for School and Child Care Centres [64]. Therefore, compared to the frequency of cleaning and sanitising, using effective agent to clean and disinfect contaminated surfaces effectively is the key to reduce viral load and prevent HFMD transmission. The association of sharing utensils with siblings and HFMD infections was not significant in this study suggest that the transmission may not necessary from saliva. However, sharing household articles such as eating utensils and toys with index case was found to be a risk factor of HFMD infections within household [19].

## Preschool admission and HFMD

Exposure to childcare centre with longer admission period, childcare centre with >115 children and larger class size with>21 children had higher risk of HFMD infections. Many studies

have shown that attending preschool is a risk factor of HFMD but the evidence is weak [21, 27, 29, 32, 33]. The demand and supply of infant care and child care centre-based services in Singapore are on the rise. Transmission of disease is likely to continue rapidly with high attack rates due to high density of susceptible children at a shared restricted space in childcare centres. Moreover, higher risk of HFMD infections in childcare centre with out-sourced cleaner may suggest that in-house cleaner is a protective factor with better cleaning quality than out-sourced. However, no information was collected on cleaning regime and detergent of childcare centres in this study. Further evaluation of the cleaning quality and disinfectant used in childcare centres may be required using laboratory methods.

## Knowledge of HFMD

HFMD knowledge among cases was better than controls. Control group had significantly poorer understanding of the signs and symptoms, incubation period, susceptible population, treatment and vaccine availability of HFMD which suggest that cases may have received these information during medical consultation and/or more motivated to read up more about HFMD. In addition, it shows that the general community who may not have infected with HFMD is likely to have poor knowledge, and hence, low awareness and motivation to practise good personal and household hygiene. More than 50% of both cases and controls were unaware of the prolonged viral shedding of HFMD, which suggest that the potential transmissibility of HFMD is significantly underestimated by parents.

## Recommendations

A few recommendations to guide HFMD prevention and control are discussed. First, the prevention of transmission among close contacts both at home and in childcare centres is crucial especially with infected siblings attending the same childcare centre. Isolation of a case is necessary but not sufficient to stop transmission from home to childcare centres and vice versa. Therefore, it is important to address the gap of effective personal and environmental hygiene practices. The major agent causing HFMD is hard to destroy. More efforts are required to inactivate enteroviruses on contaminated surfaces/articles effectively to prevent HFMD transmission. Effective hand hygiene shall be encouraged to improve hand washing with soap using the correct technique as recommended by CDC to maximise efficacy. Effective environmental hygiene include promoting the use of effective cleaning agent (diluted household bleach) to inactivate enteroviruses and disinfect contaminated surfaces/articles at home and in childcare centres. Additionally, increase public health knowledge of HFMD among all parents of preschoolers especially on the viral shedding information via educational campaigns, social media outreach, and collaborate with Early Childhood Development Agency (ECDA) to engage childcare centres.

## Implications for public health

This study supported that close contact with HFMD-infected person in household causes transmission of HFMD to occur rapidly. However, the evidence on the hygiene practices was not aligned with studies in other countries may suggest that gaps exist between knowledge and practice ofproper hand washing techniques and effective environmental cleaning methods to stop the spread of HFMD during an outbreak. Current advice from the Ministry of Health (MOH) in Singapore to the public emphasizes on the importance of maintaining high standards of personal and environmental hygiene to minimise the risk of HFMD which include frequent hand washing with soap, cover mouth and nose with a tissue when coughing or sneezing, avoid sharing food/drinks or articles with others, and disinfect contaminated toys.

However, pragmatically, the effectiveness of these hygiene practices based on practical rather than theoretical basis may not be the same as the efficacy or effectiveness shown in a well-defined controlled setting. To disinfect HFMD viruses is not an easy task due to the non-enveloped nature which is more resistant. Proper hygiene method and technique, using the right disinfecting agent are the least emphasized in current practice. Therefore, these gaps should be highlight to strengthen current prevention and control of HFMD.

This study also highlighted the importance of childcare centre environment as a risk factor of HFMD infections which include the total number of children enrolled, class size and cleaning quality by cleaner. Childcare centres with active clusters of prolonged transmission (transmission period >16 days with number of cases >10 or transmission period >16 days with attack rate >13%) were posted on MOH website. Publishing names of childcare centre to help parents be more aware and observe for HFMD symptoms if their child is attending the particular childcare centre is not likely an effective solution to reduce the high transmission rate of HFMD. Based on MOH website, it is evident that larger childcare centres with enrolment of approximately 300 children had higher attack rate. There is an urgent need to achieve a fine balance between achieving economies of scale for childcare business and opening more large (>100 students) childcare centres with large class size (>20 students), especially if effective prevention, surveillance and control measures are not established in place.

These evidence provide crucial implications to guide more effective prevention and control of HFMD in Singapore. Nevertheless, due to potential limitations of this study, further cohort study or randomised controlled trial is required in future to establish a stronger evidence-based causal relationship between these risk factors and HFMD infections. Both clinical and laboratory-confirmed cases are necessary to validate these findings. More research is also required to evaluate the effectiveness of different types of disinfectants used in household and childcare centres.

## Strengths

Retrospective case-control approach was logistically plausible and efficient for this study. Although cases were identified retrospectively, medical certificate records that have been collected for non-research purposes by childcare centres were used to identify 86.5% clinically diagnosed cases. The recruitment of controls from childcare centres represents the source population with cases to controls ratio of 1:1. This study also accounts for local environment and social factors which can be different from studies conducted in other countries.

## Limitations

There may be selection bias of the cases and controls for the analysis of childcare centre characteristics. As the number of cases and controls were not matched strictly for childcare centres, any significant proportion differences in the variables for childcare centres were taken into consideration in this study. Recall bias was inevitable as parental-reporting was used to collect children's habits and other variables of interest. Hence to minimize potential bias and to shorten the recall period, only cases diagnosed within a two-year period were recruited. The risk of recall for birth and maternal factors was minimize das 82% of respondents to the survey were mothers. Reporting of maternal factors include breastfeeding by mothers was deemed reliable even years after given birth [65]. In addition, no differential recall bias between cases and controls because they were unaware the linkage between breastfeeding and HFMD. A standardized close-ended questionnaire was administered to allow responses in the same way and avoid differential recall among cases and controls. Questionnaire was carefully

constructed with selected research questions to maximize accuracy and reliability of pre-disease exposure recall.

Due to retrospective nature of this study, virus isolation of causative strain to confirm a case of HFMD was not conducted. Only symptomatic cases based on clinical presentations were recruited. Other acute illnesses with similar manifestations such as measles and Kawasaki disease may be mistaken as HFMD. Hence there exist a possibility of non-differential misclassification due to misdiagnosed cases and potentially weakened the effect of analysis. Furthermore, a case-control study provide evidence of correlation instead of causation. Therefore, evidence may be inadequate to establish a clear temporal sequence and unable to determine causality between risk factors and HFMD infections. There were no data before 2016 to exclude the fact the controls selected were indeed true controls as this is not a cohort study and there were no serology performed to exclude this possibility. Hence, there is a possibility of differential misclassification bias that will influence the outcomes. Lastly, HFMD is likely more infectious among children in the same school as the infected case/s. Hence, there may be potential cross-infection effect, which this study is not able to account for. In fact, this potential cross-infection effect may be minimal as the controls are also selected from the same school as the cases in 1:1 ratio.

## Conclusion

Based on the hypothesis, key risk factors found include contacts with HFMD-infected siblings, longer preschool admission, and childcare centres risk factors (out-sourced cleaner, enrolment, class size) while key protective factor include age of starting solids (weaning age). This study found that child's age, started solids at ten months old and above, having HFMD-infected siblings, longer preschool admission, mothers' age of 41–50 years, two or more children in a family, household members of grandparents, parents, child and domestic helper, higher education of parents, parents who had historical HFMD episode, wash toys with soap once every two to three weeks, sanitise toys once every two to three weeks, knowledge of HFMD, out-sourced cleaner, enrolment, and class size of childcare centre were independently associated with HFMD infections in the multivariate analysis. However, gender, breastfeeding, gestational age, frequent hand washing, siblings attending same childcare centre, share utensils and play in public playgrounds were not significantly associated with HFMD infections. Future cohort study or randomized controlled trial is needed to validate these evidences systematically.

## Supporting information

**S1 File. Questionnaire in English and Chinese.**
(DOCX)

**S1 Data.**
(XLSX)

## Acknowledgments

We would like to thank the parents and senior management of the childcare centres for their support and participation in this study. We are also thankful to Singapore Children's Society & Saw Swee Hock School of Public Health of National University of Singapore (NUS) for their financial support.

## Author Contributions

**Conceptualization:** Junxiong Pang.

**Data curation:** Jo Ann Kua.

**Formal analysis:** Jo Ann Kua.

**Funding acquisition:** Jo Ann Kua.

**Investigation:** Jo Ann Kua, Junxiong Pang.

**Methodology:** Junxiong Pang.

**Project administration:** Jo Ann Kua, Junxiong Pang.

**Resources:** Junxiong Pang.

**Software:** Junxiong Pang.

**Supervision:** Junxiong Pang.

**Validation:** Junxiong Pang.

**Visualization:** Jo Ann Kua.

**Writing – original draft:** Jo Ann Kua.

**Writing – review & editing:** Jo Ann Kua, Junxiong Pang.

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
