## [Decision Letter · Decision Letter 0]

23 Jun 2020

PONE-D-19-34540

The epidemiological risk factors of hand, foot and mouth disease among children in Singapore: a retrospective case-control study.

PLOS ONE

Dear Dr. Pang,

Thank you for submitting your manuscript to PLOS ONE. After careful consideration, we feel that it has merit but does not fully meet PLOS ONE’s publication criteria as it currently stands. Therefore, we invite you to submit a revised version of the manuscript that addresses the points raised during the review process.

The Authors are expected to address all the criticisms by all Reviewers. In particular, please provide the response rate and clarify the analysis plan of the multivariable analysis (Reviewer #1), and the overall analyses (Reviewer #2). In additional to the above comments, please address,

Table 1 and 5. Please clarify if “Siblings HFMD infection history” refers to a specific period (e.g. in the previous 1 year), also for “Parents with HFMD previously”. Did you restrict the time period before the onset dates of the cases?

We look forward to receiving your revised manuscript.

Kind regards,

Eric HY Lau, Ph.D.

Academic Editor

PLOS ONE

Journal Requirements:

2. Please include additional information regarding the survey or questionnaire used in the study and ensure that you have provided sufficient details that others could replicate the analyses. For instance, if you developed a questionnaire as part of this study and it is not under a copyright more restrictive than CC-BY, please include a copy, in both the original language and English, as Supporting Information. In addition, please refer to any post-hoc corrections to correct for multiple comparisons during your statistical analyses. If these were not performed please justify the reasons. Please refer to our statistical reporting guidelines for assistance (https://journals.plos.org/plosone/s/submission-guidelines.#loc-statistical-reporting).

Additional Editor Comments (if provided):

The Authors are expected to address all the criticisms by all Reviewers. In particular, please provide the response rate and clarify the analysis plan of the multivariable analysis (Reviewer #1), and the overall analyses (Reviewer #2). In additional to the above comments, please address,

1. Table 1 and 5. Please clarify if “Siblings HFMD infection history” refers to a specific period (e.g. in the previous 1 year), also for “Parents with HFMD previously”. Did you restrict the time period before the onset dates of the cases?

Reviewers' comments:

Reviewer's Responses to Questions

**Comments to the Author**

1. Is the manuscript technically sound, and do the data support the conclusions?

Reviewer #1: Yes

Reviewer #2: Yes

2. Has the statistical analysis been performed appropriately and rigorously? 

Reviewer #1: Yes

Reviewer #2: No

3. Have the authors made all data underlying the findings in their manuscript fully available?

Reviewer #1: Yes

Reviewer #2: Yes

4. Is the manuscript presented in an intelligible fashion and written in standard English?

Reviewer #1: Yes

Reviewer #2: Yes

5. Review Comments to the Author

Reviewer #1: Based on a case-control design, the authors compared numerous personal, family and childcare center factors between the two groups (HFMD VS controls), and investigated the potential association between the risk of HFMD and these factors through univariate and multivariate logistic regressions. Both the design the statistical methods are appropriate. The manuscript was well organized and clearly written, although it’s kind of lengthy. I only have few comments:

1) “Child’s age between 1.5 and 4.9 years, had more than 1.9 years in childcare…”: is there any factor representing the “1.9 years in childcare” thing?

2) “However based on a systematic review, the meteorological effect was weak with no evidence of relationship between humidity, temperature and HFMD transmission” (Lines 85-86): as far as I know, the associations between the risk of HFMD and weather factors are well documented in literature.

3) “Based on a systematic review for risks factors of HFMD, it was suggested that age, gender, hygiene, and social contacts are associated with high risk of HFMD infections”: terms such as “age” (male or female) and “gender” are unclear. Please specify.

4) The response rate of surveys should be reported.

5) Information on 1) whether all factors were tested with the univariate logistic analysis, and 2) whether all significant ones were then included into the multivariate logistic regression should be clarified. In addition, for lines 400-402, I suggest the authors at least include a sensitivity analysis by including all significant factors identified in the univariate logistic analysis into the multivariate logistic regression.

Reviewer #2: The authors explored risk factors of hand, foot and mouth disease among children in Singapore. Generally speaking, this manuscript was well designed and well-written. Followings are some advices for reference.

1. Line 191-192: “A control was defined as a child with no history of clinically diagnosed HFMD between year 2016 and 2018”. Were the children who suffered HFMD before 2016 selected as controls? The antibody in their bodies might induce bias. Please discuss.

2. Line 194-195: “ The same year” need to be clarified. Is it the same age, or same grade, or the same research years?

3. Line 194-195: The authors stated that “Every case identified was randomly matched with a control in the same school and in the same year”. That is, the case and control were equally in different childcare centre. Then, why “Enrolment of childcare centres was significantly different among cases and controls”(Line 366)?

4. Line 182; The authors stated that “unmatched case-control study was conducted”. Then, it is contradictory to say that “Every case identified was randomly matched with a control in the same school and in the same year”(Line 194-195), which is a individual matching case-control study. For individual matching case-control study, conditional logistic regression analysis should be used, but not “Unconditional logistic regression analysis”(Line 252).

5. I wonder if this manuscript exceeded the word limitation?

6. PLOS authors have the option to publish the peer review history of their article (what does this mean?). If published, this will include your full peer review and any attached files.

Reviewer #1: No

Reviewer #2: No

---

## [Author Response · Author response to Decision Letter 0]

28 Jun 2020

Authors’ response to Editor:

1. Table 1 and 5. Please clarify if “Siblings HFMD infection history” refers to a specific period (e.g. in the previous 1 year), also for “Parents with HFMD previously”. Did you restrict the time period before the onset dates of the cases?

Authors: Thank you for your comment. The authors would like to clarify that the period of focus for “Siblings HFMD infection history” and “Parents with HFMD previously” is at the point when the questionnaire was rolled out. In order to provide clarity, the authors have added this point in the table 1 and table 5.

Authors’ response to Reviewers:

Reviewer #1: Based on a case-control design, the authors compared numerous personal, family and childcare center factors between the two groups (HFMD VS controls), and investigated the potential association between the risk of HFMD and these factors through univariate and multivariate logistic regressions. Both the design the statistical methods are appropriate. The manuscript was well organized and clearly written, although it’s kind of lengthy. 

Authors: Thank you for your comments. As there were a number of interesting findings that we think are worthwhile discussing and highlighting, the length of the manuscript is reasonably longer. 

I only have few comments:

1) “Child’s age between 1.5 and 4.9 years, had more than 1.9 years in childcare…”: is there any factor representing the “1.9 years in childcare” thing?

Authors: Thank you for the comment. The “1.9 years in childcare” represents the total duration of the child in that childcare centre at the point of the study. The study observed that the longer the duration a child had been in a childcare, there is a higher risk of having HFMD. In order to provide clarity, the following has been edited: “child who had been to childcare for more than 1.9 years…” in the abstract. 

2) “However based on a systematic review, the meteorological effect was weak with no evidence of relationship between humidity, temperature and HFMD transmission” (Lines 85-86): as far as I know, the associations between the risk of HFMD and weather factors are well documented in literature.

Authors: Thank you for the comment. Indeed there were a number of studies that had reported on the associations between risk of HFMD and climatic factors. However, from systematic and meta-reviews, including https://pubmed.ncbi.nlm.nih.gov/29306826/ , the effect of climatic factors on the risk of HFMD has been shown to be low, even though it is a significant association. In order to provide clarity, the sentence at line 85-86 has been edited to “However, based on a systematic and meta-review, the meteorological effect was weak with no evidence of strong relationship between humidity, temperature and HFMD transmission.”

3) “Based on a systematic review for risks factors of HFMD, it was suggested that age, gender, hygiene, and social contacts are associated with high risk of HFMD infections”: terms such as “age” (male or female) and “gender” are unclear. Please specify.

Authors: Thank you for the comment. In order to clarify these terms, the following have been edited in line 109: “….young age of less than five, male gender, poor hygiene and high frequency of social contacts…” In addition, the following sentences in the same section were meant to elaborate on these risk factors.

4) The response rate of surveys should be reported.

Authors: Thank you for the comment. The response rate is 33.3%. This is added in the line 266-267.

5) Information on 1) whether all factors were tested with the univariate logistic analysis, and 2) whether all significant ones were then included into the multivariate logistic regression should be clarified. In addition, for lines 400-402, I suggest the authors at least include a sensitivity analysis by including all significant factors identified in the univariate logistic analysis into the multivariate logistic regression.

Authors: Thank you for the comment. All factors were tested with the univariate logistic analysis and all significant variables were included in the initial multivariate logistic regression analysis. This has been made clearer between line 255-257. However, the more variables that were included in the model, the more likely the model will be overfitted. As a principle of choosing the most parsimonious model, variables which are not likely to be confounders, or associated with the outcome should be remove from the model. The authors felt that there is no added value to include all significant factors into the final multivariate model. 

Reviewer #2: The authors explored risk factors of hand, foot and mouth disease among children in Singapore. Generally speaking, this manuscript was well designed and well-written. Followings are some advices for reference.

1. Line 191-192: “A control was defined as a child with no history of clinically diagnosed HFMD between year 2016 and 2018”. Were the children who suffered HFMD before 2016 selected as controls? The antibody in their bodies might induce bias. Please discuss.

Authors: Thank you for the comment. The data available for this study is only between 2016 and 2018. There were no data before 2016 for us to include these children as this is not a cohort study. Agree that there is a possibility that they may have been infected before 2016, and hence, result in potential differential misclassification bias as being a control instead of a case due to immunity developed previously. This has been added as one of the limitations and discussed from line 774-777.

2. Line 194-195: “The same year” need to be clarified. Is it the same age, or same grade, or the same research years?

Authors: Thank you for the comment. The authors were referring to the same research years involved. i.e. 2016-2018. The authors have edited to make it clearer at line 197.

3. Line 194-195: The authors stated that “Every case identified was randomly matched with a control in the same school and in the same year”. That is, the case and control were equally in different childcare centre. Then, why “Enrolment of childcare centres was significantly different among cases and controls” (Line 366)?

Authors: Thank you for the comment. The term “enrolment” refers to the total number of children enrolled in the specific participating childcare centres. The authors have edited the phrase to provide clarity at line 374. The authors also noted that ““Every case identified was randomly matched with a control in the same school and in the same year” was inaccurate. This has been edited to “Cases and controls were systematically identified from all participating childcare centres who were enrolled between 2016 and 2018 for the recruitment of their parents for the questionnaire” at line 196-197.

4. Line 182; The authors stated that “unmatched case-control study was conducted”. Then, it is contradictory to say that “Every case identified was randomly matched with a control in the same school and in the same year” (Line 194-195), which is an individual matching case-control study. For individual matching case-control study, conditional logistic regression analysis should be used, but not “Unconditional logistic regression analysis” (Line 252).

Authors: Thank you for the comment. The authors noted that ““Every case identified was randomly matched with a control in the same school and in the same year” was inaccurate. This has been edited to “Cases and controls were randomly identified from all participating childcare centres who were enrolled between 2016 and 2018 for the recruitment of their parents for the questionnaire” at line 196-197. Hence, this is an unmatched case-control study. 

5. I wonder if this manuscript exceeded the word limitation.

Authors: Thank you for the comment. The authors had tried to concisely describe and discuss the results. However, as there are a number of interesting findings that we think are worthwhile discussing and highlighting, the length of the manuscript is reasonably longer. There is no restriction on word limits.

---

## [Editor Report · Decision Letter 1]

14 Jul 2020

The epidemiological risk factors of hand, foot, mouth disease among children in Singapore: a retrospective case-control study.

PONE-D-19-34540R1

Dear Dr. Pang,

We’re pleased to inform you that your manuscript has been judged scientifically suitable for publication and will be formally accepted for publication once it meets all outstanding technical requirements.

Kind regards,

Eric HY Lau, Ph.D.

Academic Editor

PLOS ONE
---

## [Editor Report · Acceptance letter]

30 Jul 2020

PONE-D-19-34540R1 

The epidemiological risk factors of hand, foot, mouth disease among children in Singapore: a retrospective case-control study. 

Dear Dr. Pang:

I'm pleased to inform you that your manuscript has been deemed suitable for publication in PLOS ONE. Congratulations! Your manuscript is now with our production department. 

Kind regards, 

on behalf of

Dr. Eric HY Lau 

Academic Editor

PLOS ONE